# Fertilizer Effects on the Nitrogen Isotope Composition of Soil and Different Leaf Locations of Potted *Camellia sinensis* over a Growing Season

**DOI:** 10.3390/plants13121628

**Published:** 2024-06-13

**Authors:** Zuchuang Guo, Chunlin Li, Xin Li, Shengzhi Shao, Karyne M. Rogers, Qingsheng Li, Da Li, Haowei Guo, Tao Huang, Yuwei Yuan

**Affiliations:** 1College of Food Sciences and Engineering, Ningbo University, Ningbo 315211, China; gzc1298848693@126.com; 2State Key Laboratory for Managing Biotic and Chemical Threats to the Quality and Safety of Agro-Products, Institute of Agro-Product Safety and Nutrition, Zhejiang Academy of Agricultural Sciences, Hangzhou 310021, China; chunlinli0304@163.com (C.L.); k.rogers@gns.cri.nz (K.M.R.); 3Key Laboratory of Tea Quality and Safety Control, Ministry of Agriculture, Tea Research Institute, Chinese Academy of Agricultural Sciences, Hangzhou 310008, China; lixin@tricaas.com; 4National Isotope Centre, GNS Science, 30 Gracefield Road, Lower Hutt 5040, New Zealand; 5Institute of Sericulture and Tea, Zhejiang Academy of Agricultural Sciences, Hangzhou 310021, China; liqs@zaas.ac.cn (Q.L.);; 6Tea Research Institute, College of Agriculture & Biotechnology, Zhejiang University, Hangzhou 310058, China

**Keywords:** tea plants, *Camellia sinensis*, fertilizer, nitrogen isotopes, leaf locations, temporal variation

## Abstract

The nitrogen-stable isotopes of plants can be used to verify the source of fertilizers, but the fertilizer uptake patterns in tea (*Camellia sinensis*) plants are unclear. In this study, potted tea plants were treated with three types of organic fertilizers (OFs), urea, and a control. The tea leaves were sampled over seven months from the top, middle, and base of the plants and analyzed for the *δ*^15^N and nitrogen content, along with the corresponding soil samples. The top tea leaves treated with the rapeseed cake OF had the highest *δ*^15^N values (up to 6.6‰), followed by the chicken manure, the cow manure, the control, and the urea fertilizer (6.5‰, 4.1‰, 2.2‰, and 0.6‰, respectively). The soil treated with cow manure had the highest *δ*^15^N values (6.0‰), followed by the chicken manure, rapeseed cake, control, and urea fertilizer (4.8‰, 4.0‰, 2.5‰, and 1.9‰, respectively). The tea leaves fertilized with rapeseed cake showed only slight *δ*^15^N value changes in autumn but increased significantly in early spring and then decreased in late spring, consistent with the delivery of a slow-release fertilizer. Meanwhile, the *δ*^15^N values of the top, middle, and basal leaves from the tea plants treated with the rapeseed cake treatment were consistently higher in early spring and lower in autumn and late spring, respectively. The urea and control samples had lower tea leaf *δ*^15^N values than the rapeseed cake-treated tea and showed a generalized decrease in the tea leaf *δ*^15^N values over time. The results clarify the temporal nitrogen patterns and isotope compositions of tea leaves treated with different fertilizer types and ensure that the *δ*^15^N tea leaf values can be used to authenticate the organic fertilizer methods across different harvest periods and leaf locations. The present results based on a pot experiment require further exploration in open agricultural soils in terms of the various potential fertilizer effects on the different variations of nitrogen isotope ratios in tea plants.

## 1. Introduction

Tea (*Camellia sinensis* (L.) Kuntze) is one of the most widely consumed non-alcoholic beverages in the world [1,2]. Tea is well-known for its bioactive compound content, with reported antioxidant, anti-cancer, anti-diabetes, and anti-obesity functions [3,4,5]. In recent years, consumers have become increasingly drawn towards organic tea products as they believe that organic tea is safer and healthier than conventional tea [6,7,8]. However, the superior market prices of organic tea relative to conventional tea increase the risk of adulteration or mislabeling [9]. Previous studies have shown that nitrogen-stable isotopes (*δ*^15^N) are useful to identify organic products from their conventional counterparts [10,11], but the temporal change in the nitrogen isotopes in tea plants by different fertilizers is still unclear.

Nitrogen-stable isotopes (*δ*^15^N) are mainly related to the nitrogen fixation patterns in plant-microbial symbiosis, soil cultivation, and fertilization methods [12,13]. They have been successfully used to identify organic crops from their conventional counterparts, such as rice, yams, and vegetables [14,15,16]. A key difference between organic and conventional crop cultivation is whether the applied nutrients are derived from organic or chemical fertilizers [17,18]. Studies have shown that the *δ*^15^N values of chemical fertilizers are close to air (~0‰), while the *δ*^15^N values of organic fertilizers from different sources are between +0.6‰ and 16.2‰ [19,20,21]. The fertilizers commonly applied in China during organic tea cultivation include organic plant fertilizers such as rapeseed cake, and animal wastes such as chicken, pig, or cow manure. Previous studies have used different fertilizer types (including fish meal, rapeseed cake, and chemical fertilizer) to grow tea plants and found that the nitrogen isotope differences among the tea leaves correlated with the planting year and tea variety [22]. There are only a few tea studies that investigate the nitrogen isotope effects from the commonly used fertilizers in China.

Plants assimilate, metabolize, and reflect the nitrogen sources from fertilizers. Different parts of the plant can also vary in N content and ^15^N abundance according to the *δ*^15^N values of the applied fertilizer [23]. The previous studies found that the newly formed leaves on sweet pepper plants had the highest *δ*^15^N values, followed by the stems and older leaves, and the lowest values were found in the roots [24]. A few studies discuss the temporal isotopic changes for different parts of tea plants. A previous study showed that the *δ*^15^N values of the young tea leaves had more positive *δ*^15^N values than the older leaves [25]. Tea is usually harvested in spring, summer, and autumn, and the nitrogen isotopes of tea plants at different collecting periods are also different. Other studies found that, for field-grown tea plants, the *δ*^15^N values of the young leaves picked on 20 March 2023 (3.6‰) had lower *δ*^15^N values than the young leaves picked on 30 May 2023 (4.2‰) [26]. Accordingly, these dynamic *δ*^15^N leaf value changes across different tea collection periods need to be studied in more detail. Most of the previous nitrogen isotope tea studies consist of field trials conducted in tea gardens that are traditionally perched on steep slopes, with highly variable nitrogen leaching, surface runoff, and nitrogen-fixing microbial activities that affect the nutrient input to the tea plant [27]. Researchers have reported that the plant *δ*^15^N values in plant–soil systems are closely related to the soil nitrogen content, nitrogen cycling rate, and plant nitrogen uptake strategy [28]. It is necessary to design pot experimental conditions to minimize these external influences, ensuring that only the fertilizer applied to the tea plant is the main nitrogen isotope variable.

In this study, organic fertilizers (OFs, including rapeseed cake, chicken manure, and cow manure), a chemical fertilizer (CF, urea), and no fertilizer (control) were used to carry out pot experiments on tea plants. The nitrogen isotopes were determined for the tea leaves in different spatial positions on the plant during different collecting periods. This study contributes towards understanding the temporal dynamic changes in the nitrogen-stable isotopes in tea plants from different fertilizer types and provides a fundamental basis for the study of the nitrogen isotope composition and transfer mechanism for organic tea.

## 2. Materials and Methods

### 2.1. Pot Experiments and Sampling

Three-year-old potted tea plants (Longjing 43 variety) were selected for experimental fertilizer trials at the Tea Research Institute, Chinese Academy of Agricultural Sciences, China (latitude 30°17′99.62″ N; longitude 120°09′39.20″ E) at an altitude of 16 m. Five different fertilizer treatments were selected: organic fertilizers (rapeseed cake, chicken manure, and cow manure), a chemical fertilizer (urea), and a control (no fertilizer). Each fertilizer treatment had 3 replicate pots, with a total of 15 pots containing a single tea plant around 60 cm tall. The size of each experimental pot was 20 cm in diameter at the base, 22 cm in diameter at the top, and 20 cm in height and contained around 3 kg of soil in each pot. The initial %N and *δ*^15^N values of the tea plants and soil are represented by the control samples. Potted tea plants were fertilized only once at the start of the study (5 October 2022), and cultivation conditions (temperature, irrigation water, humidity, sunshine hours, etc.) were the same as those used by local tea gardens except for the different fertilizer treatments. Fertilizer type, nitrogen content, dosage, and *δ*^15^N values used in the experiment are shown in Table 1, with around 1.6 g of nitrogen delivered to each pot at the start of the experiment, apart from the control samples.

Eight sampling events from each pot were undertaken during the study, covering the period of autumn through to late spring growth. In Hangzhou, tea leaf samples from the same growth phase collected on 20 October and 18 November 2022 were classified as autumn; 17 March, 21 April, and 10 May 2023 as early spring; and 20 May and 30 May 2023 as late spring. Top leaves were collected from the top 10 cm of the plant, middle leaves were collected between 20 and 40 cm above the soil, and basal leaves were collected between 0 and 20 cm above the soil. Around 2–3 g of soil and tea leaves were collected from each pot during each sampling event. Soil samples were air-dried at 25 °C and then ground into fine powder (<100 mesh). The freshly picked tea leaves were vacuum-dried for 48 h, crushed and finely ground (<80 mesh, with an aperture of 0.180 mm), and stored over silica gel in a desiccator prior to analysis. Experimental design of the tea plants is shown in Figure 1.

### 2.2. Stable Isotope Analysis

Nitrogen content (%N) and isotope values (*δ*^15^N) of tea leaves and soil samples were determined using an elemental analyzer (Isotope PYRO cube, Elementar, Langenselbold, Germany) coupled with an isotope ratio mass spectrometer (Isoprime 100, Elementar, Handforth, UK). Multi-point isotope calibration was applied for more accurate measurement using primary reference standard materials provided by IAEA (International Atomic Energy Agency, Vienna, Austria), including B2155 (protein, *δ*^15^N = 5.94‰), USGS40 (L-glutamic acid, *δ*^15^N = −4.52‰), USGS64 (glycine, *δ*^15^N = 1.76‰), and IAEA-N_2_ ((NH_4_)_2_SO_4_, *δ*^15^N = 20.3‰). High-purity N_2_ gas served as a reference gas, and helium (He) gas was used as the carrier gas during the analytical process. Elemental analyzer oxidation and reduction furnace temperatures were set at 1020 °C and 650 °C, respectively.

Tea leaves (4 mg) and soil (10 mg) were weighed in duplicate (Mettler Toledo, XP6, d = 1 μg, Greifensee, Switzerland) and packed into 6 × 4 mm^2^ tin capsules for *δ*^15^N analysis. Stable isotope ratios are expressed by the following Equation (1):*δ*X = (R_sample_ − R_reference_)/R_reference_(1)
where *δ*X is the *d*^15^N value, and R_sample_ and R_standard_ denote the abundance of the ‘heavy’ to ‘light’ isotope ratio of unknown samples and reference materials [29]. The analytical precision and reproducibility are ±0.2‰ for *δ*^15^N.

The formulas used to calculate the isotope composition differences between leaves from different parts of the tea plant were as follows in Equation (2):Δ^15^N_basal-middle_ = *δ*^15^N_middle_ − *δ*^15^N_basal_ or Δ^15^N_middle-top_ = *δ*^15^N_top_ − *δ* ^15^N_middle_(2)
where *δ*^15^N values are from the basal, middle, and top leaves and Δ^15^N is the difference between two locations.

### 2.3. Statistical Analysis

One-way analysis of variance (ANOVA) was performed using SPSS 22.0 software (IBM, Armonk, NY, USA) to calculate the mean and standard deviation (SD) of nitrogen isotope values (*δ*^15^N) for leaves of different positions and soil of tea plants. Least significant difference method (LSD) was used to evaluate significant differences between different sets of data, where a significant difference is defined as *p* < 0.05. Origin 2019b (OriginLab, Northampton, MA, USA) software was used for mapping the sample locations.

## 3. Results and Discussion

### 3.1. Fertilizer Effect on Nitrogen Content and δ^15^N Values of Top Tea Leaves and Soil

A comparison of the top tea leaf %N and *δ*^15^N values was conducted after different fertilizer treatments, and the results are shown in Appendix A and Table 2 and Figure 2. Initially, the nitrogen content of the top leaves was similar for all the fertilizer treatments and ranged between 2.5 and 3.0%. During the cooler autumn months (October to November 2022), the N content rose slightly as the tea plant was more dormant, and it stored nitrogen in the leaves. As early spring arrived, a rapid increase was noted in the %N due to the rapid growth of early spring leaves (March to 10 May 2023), and then a further %N decrease (potentially due to cold weather) occurred during late spring (20 May to 30 May 2023), and then, as the weather warmed, the leaf growth resumed.

Overall, the N content of the tea leaves was similar under different fertilizer treatments. Initially, in autumn, the organically fertilized top tea leaves had slightly lower N content than the tea leaves fertilized using urea or the control pots. The nitrogen from the organic fertilizers is not as chemically available to the plant as the soluble urea fertilizer, but, over the duration of the study, the N content of the urea-treated tea plants decreased more rapidly than that of the organic-treated plants. The organic fertilizers appear to provide a more sustained and continuous delivery of plant nutrients over time compared to chemical fertilizers.

The *δ*^15^N values of the top leaves treated with the OFs (chicken manure, rapeseed cake, and cow manure) from each sampling period were significantly more positive (3.4‰ to 6.6‰) than those treated with the CF (−0.9‰ to 0.6‰) and the control samples (0.3‰ to 2.2‰). The *δ*^15^N values of the top leaves treated with the CF had the lowest *δ*^15^N values of all five treatments. On 20 October and 18 November 2022 and 17 February, 17 March, and 21 April 2023, the *δ*^15^N values of the top leaves treated with chicken manure had the highest *δ*^15^N values, ranging from 5.4‰ to 6.5‰. However, on 10, 20, and 30 May 2023, the *δ*^15^N values of the top leaves treated with the rapeseed cake were higher and ranged from 5.2‰ to 6.6‰. The application of the OF systematically increased the *δ*^15^N values of the tea leaves, similar to previous studies of other crops [30,31]. OF is derived from organically grown plant debris or animal waste, providing a comprehensive fertilizer for tea plants with a longer-term nutrient release effect [32,33]. Meanwhile, OFs can improve the net photosynthetic rate, stomatal conductance, transpiration rate, and other photosynthetic characteristics of tea plants, increasing the dry matter accumulation and the *δ*^15^N values of tea leaves [34].

The urea application resulted in a decrease in the tea leaves’ *δ*^15^N values, which may be related to the preferential discrimination of ^15^N by nitrate reductase and glutamine synthetase during nutrient uptake and assimilation, as well as the factors including the nutrients provided by the urea being rapidly absorbed by the tea plants, nitrogen leaching, and poor microbial activity [35,36,37]. While the OF *δ*^15^N values of the chicken manure and cattle manure were similar to each other, and significantly higher than those of the rapeseed cake (Table 1), the resulting *δ*^15^N values of the tea leaves treated with chicken manure and rapeseed cake were higher than those of the tea plants treated with cow manure (Table 2). This may be attributed to the urea or ammonia content found in fresh manure, which may cause a temporary decrease in tea leaf *δ*^15^N values due to mineralization. Cow manure (as a fertilizer) is also prone to nitrogen leaching and results in lower microbial activity than chicken manure or rapeseed cake fertilizer, resulting in lower *δ*^15^N values. Previous studies have also shown that the content of ammonia nitrogen in cow manure is low, and tea trees preferentially absorb ammonia nitrogen [38,39]. In previous ecology studies, some papers reported that the soil nitrogen composition is the main factor affecting plant *δ*^15^N values, and the plant leaf *δ*^15^N values are strongly related to the nitrogen absorption strategies regarding different nitrogen forms [40,41].

Different pot fertilizer treatments not only changed the nitrogen content and isotope characteristics of the tea leaves but also those of the pot soil (Appendix A and Table 3). The soil nitrogen content slowly increased in autumn (October to November 2022) for the OF (1.4% to 2.2%), CF (0.8% to 1.4%), and control treatments (1.3% to 1.4%). Although no further fertilizer was applied to the soil in early spring (March to 10 May 2023), the soil N content also slowly increased for the OF (1.4% to 2.3%) and control treatments (1.1% to 1.3%). All the treatment soils demonstrated a gradual decrease in %N during late spring (20 May to 30 May 2023). The soil values show similar trends to the tea leaves’ %N, suggesting that warmer spring conditions and root exudates may be conducive to symbiotic N-producing soil microbes [42].

The soil *δ*^15^N values showed similar trends to the *δ*^15^N values of the top leaves, where the highest *δ*^15^N values were found in the OF-treated soils and the lowest *δ*^15^N values in the CF-treated soils. The *δ*^15^N values of the soil treated with the OF (3.3‰ to 6.0‰) became significantly more positive than the control soil (2.0‰ to 2.5‰), and the soils treated with the CF (1.2‰ to 1.9‰) became less positive than the control soil. During the study period, the cattle manure-treated soil *δ*^15^N sampled on November 18 had the highest soil *δ*^15^N value (6.0‰), and the CF urea-treated soil sampled on 30 May had the lowest *δ*^15^N value (1.2‰). The OF was shown to effectively improve the activity of the soil microorganisms and soil enzymes and can promote the growth and reproduction of microorganisms, enabling more nitrogen to be assimilated, absorbed, and utilized by the soil microorganisms, thereby increasing the soil nitrogen storage [43,44]. Some previous studies have shown that CF urea can lead to soil compaction acidification and a damaged soil ecological environment [45]. Among the three organic fertilizer treatments, the cow manure-treated soils had the highest *δ*^15^N values, followed by the chicken manure-treated soils, and the rapeseed cake-treated soils had the lowest *δ*^15^N values. This trend is most likely due to the rapid denitrification of animal manure relative to the degradation of plant organic matter, resulting in higher soil *δ*^15^N values [46].

### 3.2. Temporal δ^15^N Variations in Tea Leaves Treated with Different Fertilizers

The temporal *δ*^15^N variations in the tea leaves (sampled from the top, middle, and basal leaves) treated with rapeseed cake, urea, and the control treatments are shown in Figure 2A, Figure 2B, and Figure 2C, respectively. The *δ*^15^N trends changed according to the different types of fertilizer applied. Rapeseed cake is the most widely used organic fertilizer in Chinese tea gardens, so this study provides a deeper investigation of the temporal effects from this fertilizer instead of the traditional animal manures. The control samples (no fertilizer added) equate to the tea plants growing in their natural state without fertilizer treatment. The *δ*^15^N values of the control top, middle, and basal leaves show an overall temporal downward trend over the study period, ranging from 2.6‰ to 0.2‰, although there was a slight increase from 17 March to 10 May (Figure 2C). The *δ*^15^N values of the top, middle, and basal tea leaves treated with rapeseed cake showed minor fluctuations between 20 October and 17 February (4.8‰ to 5.6‰) and a significant increase from 17 March to 10 May (4.8‰ to 6.6‰) until 30 May, and then the *δ*^15^N values rapidly decreased to 5.1‰ (Figure 2A). The urea treatment showed an upward *δ*^15^N trend from 20 October to 18 November and from 17 March to 10 May, and a downward trend for the rest of the study period (Figure 2B).

Overall, the temporal tea leaf *δ*^15^N changes for the three fertilizer treatments appear to have a strong association with the temperature and annual growth patterns of the tea plants in this region. The *δ*^15^N values of the top, middle, and basal leaves in autumn showed minor fluctuations as the temperature slowly decreased in autumn. At this time, the tea plants gradually entered a metabolic dormancy state, and the leaf nitrogen metabolism was relatively slow. During the dormancy state, most of the photosynthetic products (such as proteins) produced by leaf photosynthesis are stored in the rhizomes [47]. The *δ*^15^N values of the top, middle, and basal leaves increased in early spring as the warmer temperatures were more suitable for photosynthesis [48]. During early spring, most of the photosynthetic products were transferred to the axillary buds, while the nutrients stored in the roots and stems were also rapidly remobilized and transferred to the buds, encouraging new leaf growth [49]. This sharp nitrogen metabolic reaction in the leaves results in an increase in the *δ*^15^N values. Finally, the *δ*^15^N values of the top, middle, and basal leaves decreased in late spring, mainly due to a reduction in the leaf photosynthetic efficiency with increasing temperatures as the tea plants entered the summer metabolic resting period. The top leaf *δ*^15^N values (red line) were generally more positive than the basal leaves (blue line) for the different treatments and sampling events. This may be due to a higher ^15^N fractionation effect caused by more light exposure, photosynthesis, and growth within the top leaves compared to the basal leaves. Plant studies have shown that the lighter ^14^N isotope will be preferentially metabolized (fractionated) and allocated under higher photosynthetic rates, thus enriching ^15^N [50].

### 3.3. δ^15^ N Differences between the Top, Middle, and Basal Tea Leaves

The temporal tea leaf *δ* ^15^N differences (Δ^15^N) between the top, middle, and basal leaf positions under three fertilizer treatments (rapeseed cake, urea, and control treatments) are shown in Table 4. The results show that, during autumn and early spring, the tea leaves treated with rapeseed cake showed a *δ*^15^N pattern similar to the control samples: an ^15^N enrichment from the basal to middle leaves and a depletion in ^15^N from the middle to top leaves. The tea plants treated with CF urea showed an opposite trend, with a depletion from the basal to middle leaves and then an enrichment from the middle to top leaves. In autumn, the Δ^15^N isotopic differences between the basal to middle and middle to top (*δ*^15^N_basal-middle_ and *δ*^15^N_middle-top_) of the potted tea plants treated with rapeseed cake were 0.6‰ and −0.2‰, the control treatments were 0.5‰ and −0.3‰, and the urea treatments were −0.5‰ and 1.0‰. In early spring, the Δ^15^N isotopic differences between the basal to middle and middle to top of potted tea plants treated with rapeseed cake were 0.7‰ and −0.1‰, the control treatments were 0.2‰ and −0.2‰, and the urea treatments were −0.6‰ and 0.8‰. In late spring, the tea plants treated with rapeseed cake showed isotopic enrichment from the basal to middle leaves and depletion from the middle to top leaves. The tea plants treated with urea and the control samples were enriched in ^15^N from the basal to middle leaves and middle to top leaves, with Δ^15^N isotopic differences for the urea treatment of 0.6‰ and 0.1‰, and the control differences were 0.3‰ and 0.3‰.

The tea leaf *δ*^15^N isotopic values are associated with the metabolism and transport efficiency of nitrogen to different tissue sites in tea plants. The roots absorb nitrogen from the soil and convert it into amino acids, which the tea plant requires for growth [51]. When tea plants grow rapidly, e.g., in spring, amino acids are transported through the stem to the leaves to provide essential nutrients for ongoing cell division and multiplication, which is accompanied by nitrogen isotope fractionation [52,53]. The Δ^15^N isotopic differences from the basal to middle and top leaves of the control plants first show enrichment and then depletion in autumn and early spring, and continuous enrichment in late spring. The results showed that the nitrogen metabolism of the control plants was more vigorous in the middle leaves than in the basal and top leaves in autumn and early spring, while the nitrogen metabolism of the top leaves was more vigorous in late spring.

The Δ^15^N isotopic leaf position differences of the potted tea plants treated with rapeseed cake were the same across all the seasons. Firstly, the tea plants were enriched from the basal to middle leaves and then depleted from the middle to top leaves. Rapeseed cake continuously provides nitrogen nutrients to tea plants from autumn to late spring, providing a sustained long-term effect on the nitrogen isotope metabolism of tea plants. The middle leaves from each sampling period treated with the rapeseed cake fertilizer showed strong nitrogen metabolic activities, such as amino acid synthesis and transport [54]. The Δ^15^N isotopic leaf position differences of the tea plants treated with urea and the control treatment showed an opposite trend in autumn and early spring and the same trend in late spring. It is speculated that urea changed the nitrogen metabolism of tea plants in autumn and early spring and that urea is absorbed by tea plants in late spring [36].

## 4. Conclusions

This study demonstrates that the OF significantly increases the *δ*^15^N values of potted tea plants compared with the CF and control treatments. Compared to the urea and control treatments, the *δ*^15^N values of the basal, middle, and top leaves sampled from the rapeseed cake-treated tea plants showed a significant increase in early spring. The application of the rapeseed cake fertilizer changed the *δ*^15^N values of the tea leaves sampled from different locations on the plant, first showing enrichment from the basal to middle leaves and then depletion from the middle to top leaves in autumn, early spring, and late spring. Δ^15^N isotopic differences were noted between the basal, middle, and top leaves and were clearly affected by the fertilizer type and sampling period. Overall, the *δ*^15^N signatures of the OF- and CF-treated tea plants were sustained during the early and late spring harvest period, providing clear evidence that the *δ*^15^N values of the harvested tea from different locations on the tea plant can be directly related to the fertilizer type and farming methods. The outcomes of this research show that the top and middle tea leaves typically collected during a tea harvest can be reliably used to authenticate organic production claims, increasing the tea production verification tools and consumer confidence. While our research does not identify the precise internal mechanisms that give rise to these results, further studies using labelled fertilizers will be undertaken to establish the kinetic mechanisms and confirm our observations in more detail. Meanwhile, this study is only a report from a pot experiment for variations in the nitrogen isotope ratios in tea plants and soil and may not accurately reflect the fertilizer effect on the nitrogen isotope composition in open agricultural soils.

## Figures and Tables

**Figure 1 plants-13-01628-f001:**
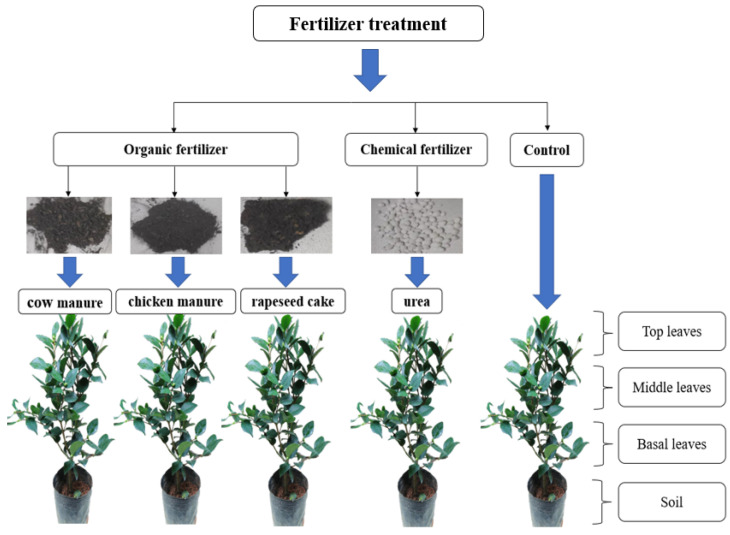
Experimental design of the tea plant study.

**Figure 2 plants-13-01628-f002:**
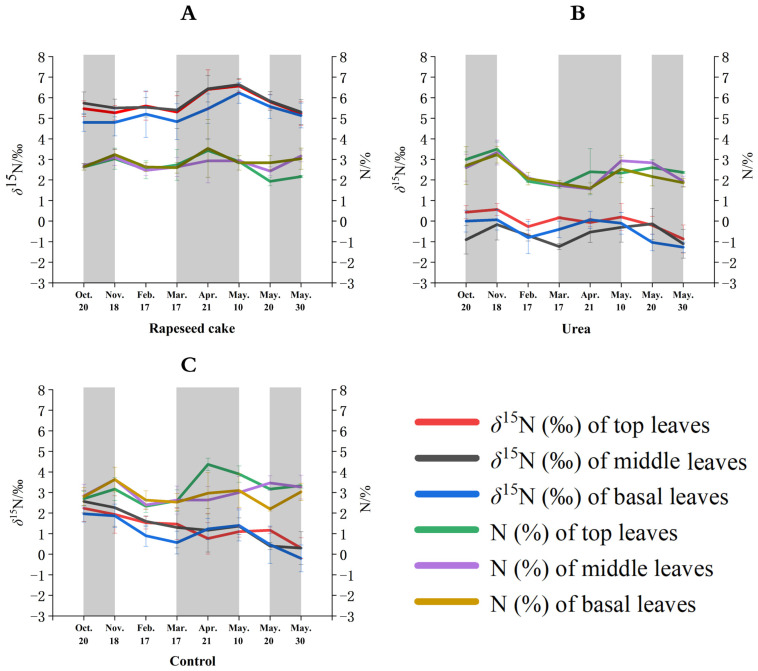
Temporal %N and *δ*^15^N values of different leaves on tea plants using (**A**) rapeseed cake, (**B**) urea, and (**C**) no fertilizer (control) treatments.

**Table 1 plants-13-01628-t001:** Nitrogen content, fertilizer dosage, and *δ*^15^N values of fertilizers.

Fertilizer Type	Name	Nitrogen Content (%)	Fertilizer Dose/Pot (g)	Nitrogen Dose/Pot (g)	*δ*^15^N Value (‰)
OF	Chicken manure	2.3	69.6	1.6	6.5
	Rapeseed cake	7.8	20.5	1.6	1.9
	Cow manure	1.5	106.7	1.6	6.9
CF	Urea	44.8	3.6	1.6	−0.6

**Table 2 plants-13-01628-t002:** Nitrogen-stable isotopes (*δ*^15^N) of top tea leaves treated with different fertilizers.

Treatment		*δ*^15^N (‰) Mean ± SD							
		2022 20 October	2022 18 November	2023 17 February	2023 17 March	2023 21 April	2023 10 May	2023 20 May	2023 30 May
OF	Chicken manure	6.5 ± 0.7 a	6.5 ± 0.7 a	6.0 ± 0.3 a	5.4 ± 0.6 a	6.5 ± 0.3 a	6.2 ± 0.2 a	5.1 ± 0.1 a	4.9 ± 0.1 ab
	Rapeseed cake	5.5 ± 0.4 b	5.3 ± 0.4 b	5.6 ± 0.7 a	5.3 ± 0.8 a	6.4 ± 1.0 a	6.6 ± 0.3 a	5.8 ± 0.3 a	5.2 ± 0.6 a
	Cow manure	3.7 ± 0.4 c	3.6 ± 0.4 c	3.5 ± 0.9 b	3.4 ± 0.6 b	3.7 ± 0.9 b	4.1 ± 0.7 b	3.4 ± 0.6 b	3.7 ± 1.1 b
CF	Urea	0.4 ± 0.3 d	0.6 ± 0.3 d	−0.3 ± 0.4 c	0.2 ± 0.1 c	−0.1 ± 0.2 c	0.2 ± 0.7 c	−0.2 ± 0.4 c	−0.9 ± 0.7 c
Control	No fertilizer	2.2 ± 0.6 e	1.9 ± 0.9 e	1.5 ± 0.3 d	1.5 ± 0.8 d	0.8 ± 0.8 c	1.1 ± 0.3 d	1.2 ± 0.2 d	0.3 ± 0.5 c

Note: lowercase letters “a, b, c, d, e” indicate significant differences between fertilizers (*p* < 0.05).

**Table 3 plants-13-01628-t003:** Nitrogen-stable isotopes (*δ*^15^N) of soils treated with different fertilizers.

Treatment		*δ*^15^N (‰) Mean ± SD							
		2022 20 October	2022 18 November	2023 17 February	2023 17 March	2023 21 April	2023 10 May	2023 20 May	2023 30 May
OF	Chicken manure	4.6 ± 0.2 a	4.8 ± 0.2 b	4.4 ± 0.1 a	4.4 ± 0.2 b	4.8 ± 0.2 a	4.5 ± 0.1 ab	4.6 ± 0.1 a	4.5 ± 0.2 b
	Rapeseed cake	3.9 ± 0.4 a	4.0 ± 0.3 c	3.3 ± 0.2 b	3.4 ± 0.6 c	3.4 ± 0.1 b	3.3 ± 0.4 bc	3.3 ± 0.3 b	3.6 ± 0.2 c
	Cow manure	4.9 ± 1.0 a	6.0 ± 0.3 a	5.0 ± 0.9 a	5.4 ± 0.7 a	5.4 ± 0.4 a	4.9 ± 1.4 a	4.6 ± 1.1 a	5.3 ± 0.4 a
CF	Urea	1.5 ± 0.6 b	1.3 ± 0.3 d	1.8 ± 0.6 c	1.6 ± 0.5 d	1.6 ± 0.7 c	1.6 ± 0.4 d	1.9 ± 0.7 c	1.2 ± 0.2 e
Control	No fertilizer	2.2 ± 0.3 b	2.0 ± 0.4 e	2.5 ± 0.2 bc	2.5 ± 0.5 cd	2.2 ± 0.2 c	2.4 ± 0.2 cd	2.3 ± 0.4 bc	2.1 ± 0.3 d

Note: lowercase letters “a, b, c, d, e” indicate significant differences between fertilizers (*p* < 0.05).

**Table 4 plants-13-01628-t004:** Δ^15^N differences between top, middle, and basal leaves during different seasons.

Fertilizer	Sampling Positions	Autumn			Early Spring			Late Spring		
		*δ*^15^N (‰)		Δ^15^N	*δ*^15^N (‰)		Δ^15^N	*δ*^15^N (‰)		Δ^15^N
		Average	SD		Average	SD		Average	SD	
Rapeseed cake	Top	5.4	0.3	-	6.1	0.9	-	5.5	0.5	-
	Middle	5.6	0.5	−0.2	6.2	0.8	−0.1	5.6	0.6	−0.1
	Basal	4.8	0.5	0.6	5.5	0.9	0.7	5.4	0.6	0.2
Urea	Top	0.5	0.3	-	0.1	0.4	-	−0.5	0.6	-
	Middle	−0.5	0.8	1.0	−0.7	0.6	0.8	−0.6	0.8	0.1
	Basal	0	0.5	−0.5	−0.1	0.4	−0.6	−1.2	0.3	0.6
Control	Top	2.1	0.7	-	1.1	0.6	-	0.7	0.6	-
	Middle	2.4	0.7	−0.3	1.3	0.8	−0.2	0.4	0.5	0.3
	Basal	1.9	0.4	0.5	1.1	0.7	0.2	0.1	0.8	0.3

## Data Availability

Data are contained within the article and Appendix A.

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
