# Peer review of "Fertilizer Effects on the Nitrogen Isotope Composition of Soil and Different Leaf Locations of Potted Camellia sinensis over a Growing Season"

_plants, 2024, doi:10.3390/plants13121628_

Round 1

Reviewer 1 Report

Comments and Suggestions for Authors

The article is well structured and written, it was a pleasure to read it

Author Response

Reviewer 1:

The article is well structured and written, it was a pleasure to read it.

Response: Thank you for your high evaluation of us, and we will continue to study isotopic changes in tea plants.

Reviewer 2 Report

Comments and Suggestions for Authors

The reviewed article is very interesting and should be published. Below are some minor comments and suggestions.

Keywords The phrase “tea plants” is in title , maybe better replaced with “Camellia sinensis”.

Line 31 Not camellia sinensis but Camellia sinensis (L.) Kuntze

The methodology does not provide information whether the collected leaves were in the same growth phase.

Perhaps it would be good to provide the total nitrogen content in the tested leaves.

The scale in the charts, Figure 2, should be the same. It is a pity that these analyzes were not presented for at least one manure.

Line 241 the word “first” is unnecessary.

Reviewer 3 Report

Comments and Suggestions for Authors

Z. Guo et al. analyzed the natural 15N compositions (δ15N) in three leaf parts and soils from potted tea plants treated with organic fertilizers, urea and no fertilizers. Based on the δ15N value differences among plant parts and soils, they further liked to discuss nitrogen isotope fractionations in plants and soils during the tea growing (from autumn to spring). However, the discussions for nitrogen isotope physical and chemical isotope fractionations in the report are completely lacking in theoretical explanations, experimental evidences, data significance, and important references to identify the sites and nature of isotopic fractionations.  

The reviewer likes to show important comments:

Comment 1 [title]: As state above, ‘fractionation’ in the title should not be included. The report title can be only “Changes of nitrogen isotope compositions during autumn and spring in three leaf parts and soils from the potted tea plants treated with organic fertilizers and urea”. There are already many reports on the isotopic compositions depending on N sources, including the high δ15N values in organic fertilized treatments.

Comment 2 [Abstract]: State this is ‘pot experiment’ but not field trials.

Comment 3 [lines 26, 39, 74 and Introduction]: Remove ‘fractionation of nitrogen stable isotopes’. No isotope fractionations in physical (transport/diffusion) and enzyme-involved reactions in soil and plants were described in Introduction. To study the isotope fractionation from root to leaves, for example, δ15N values of xylem sap are useful information. To study the isotope fractionation in nitrogen mineralization in soils, for example,δ15N values of nitrate and ammonium produced in soils are useful.

Comment 4 [lines 49, 51-67]: Not ‘few studies’ but moderate numbers of studies were reported in Asian tea-growing countries, not only China but more Japan and India. For isotopic fractionations in plant-soil systems, there are many excellent literatures.

Comment 5 [lines 77-88]: How much of soils were used in each pot? What are their nitrogen contents andδ15N values? What are theδ15N values of the tea leaves before fertilizer treatments?

Comment 6 [Table 6]: The amounts of fertilizer nitrogen dosed should be reported but not dosed fertilizer amounts.

Comment 7 [line 122]: ‘isotope fractionation differences’ here is wrong use, only ‘isotope composition differences’, which do not say anything about isotopic fractionation. Isotope fractionations occur during physical and enzyme-involved reactions.

Comment 8 [Results and discussion]: The discussions for isotope fractionations in the plants were only speculations without experimental evidences and important references by scientists working for isotopic fractionations in plants and soils. More information on the time-dependent mineralization from fertilizers and soils as well as δ15N values of mineralized N at plant N uptake is required. The discussions for Table 4 with large SD are over-discussion.

Z. Guo et al. analyzed the natural 15N compositions (δ15N) in three leaf parts and soils from potted tea plants treated with organic fertilizers, urea and no fertilizers. Based on the δ15N value differences among plant parts and soils, they further liked to discuss nitrogen isotope fractionations in plants and soils during the tea growing (from autumn to spring). However, the discussions for nitrogen isotope physical and chemical isotope fractionations in the report are completely lacking in theoretical explanations, experimental evidences, data significance, and important references to identify the sites and nature of isotopic fractionations.  

The reviewer likes to show important comments:

Comment 1 [title]: As state above, ‘fractionation’ in the title should not be included. The report title can be only “Changes of nitrogen isotope compositions during autumn and spring in three leaf parts and soils from the potted tea plants treated with organic fertilizers and urea”. There are already many reports on the isotopic compositions depending on N sources, including the high δ15N values in organic fertilized treatments.

Comment 2 [Abstract]: State this is ‘pot experiment’ but not field trials.

Comment 3 [lines 26, 39, 74 and Introduction]: Remove ‘fractionation of nitrogen stable isotopes’. No isotope fractionations in physical (transport/diffusion) and enzyme-involved reactions in soil and plants were described in Introduction. To study the isotope fractionation from root to leaves, for example, δ15N values of xylem sap are useful information. To study the isotope fractionation in nitrogen mineralization in soils, for example,δ15N values of nitrate and ammonium produced in soils are useful.

Comment 4 [lines 49, 51-67]: Not ‘few studies’ but moderate numbers of studies were reported in Asian tea-growing countries, not only China but more Japan and India. For isotoic fractionations in plant-soil systems, there are many excellent literatures.

Comment 5 [lines 77-88]: How much of soils were used in each pot? What are their nitrogen contents andδ15N values? What are theδ15N values of the tea leaves before fertilizer treatments?

Comment 6 [Table 6]: The amounts of fertilizer nitrogen dosed should be reported but not dosed fertilizer amounts.

Comment 7 [line 122]: ‘isotope fractionation differences’ here is wrong use, only ‘isotope composition differences’, which do not say anything about isotopic fractionation. Isotope fractionations occur during physical and enzyme-involved reactions.

Comment 8 [Results and discussion]: The discussions for isotope fractionations in the plants were only speculations without experimental evidences and important references by scientists working for isotopic fractionations in plants and soils. More information on the time-dependent mineralization from fertilizers and soils as well as δ15N values of mineralized N at plant N uptake is required. The discussions for Table 4 with large SD are over-discussion.

Author Response

Responses to Reviewers

Dear Editors and Reviewers,

Thanks for giving us a chance to revise the manuscript. We have carefully studied the reviewers’ comments and revised the manuscript to the best of our ability according to the comments. Below are our responses to the reviewers’ comments. The changes are heighted in red in the revised manuscript. We have also carefully gone through the manuscript again using a native English speaker to check the clarity and syntax of the text. We hope the manuscript now meets your requirements and look forward to working with you to publish our paper.

------------------------------------------------

  1. Guo et al. analyzed the natural 15N compositions (δ15N) in three leaf parts and soils from potted tea plants treated with organic fertilizers, urea and no fertilizers. Based on the δ15N value differences among plant parts and soils, they further liked to discuss nitrogen isotope fractionations in plants and soils during the tea growing (from autumn to spring). However, the discussions for nitrogen isotope physical and chemical isotope fractionations in the report are completely lacking in theoretical explanations, experimental evidences, data significance, and important references to identify the sites and nature of isotopic fractionations.  

The reviewer likes to show important comments:

Comment 1 [title]: As state above, ‘fractionation’ in the title should not be included. The report title can be only “Changes of nitrogen isotope compositions during autumn and spring in three leaf parts and soils from the potted tea plants treated with organic fertilizers and urea”. There are already many reports on the isotopic compositions depending on N sources, including the high δ15N values in organic fertilized treatments.

Response: Thanks for your constructive suggestion. We have modified the title according to your suggestion and that of Reviewer 1.

Line 2-3:

‘Fertilizer spatio-temporal effects on the nitrogen isotope composition of soil and different Camellia sinensis leaf locations’

There are already many reports on the isotopic compositions depending on N sources, including the high δ15N values in organic fertilized treatments.

Response: We acknowledge that previously many reports have been made on the application of organic fertilizers to crops/plants and have documented the resultant higher δ15N values. However, this paper focuses on different parts of the tea plant to determine which parts are most affected by fertilizers, and whether the fertilizer signature is sustained over time. Organically fertilized tea should show a higherδ15N value in the leaf, compared to conventionally fertilized counterparts. This signature should be sustained over the entire harvest period to enable authentication of farming method claims, as tea is harvested at different time periods (ie. early/late harvest). It is important to understand changes that may occur in leaves picked across the entire harvest period to assure the farming methods, tea quality and consequent production claims (ie. organic).  

Comment 2 [Abstract]: State this is ‘pot experiment’ but not field trials.

Response: Thanks for your constructive suggestion. We have emphasized that this is a pot experiment in the abstract and removed any reference to field trials.

Line 19:

In this study, potted tea plants were treated with 3 types of organic fertilizers (OF), urea and control.

Comment 3 [lines 26, 39, 74 and Introduction]: Remove ‘fractionation of nitrogen stable isotopes’. No isotope fractionations in physical (transport/diffusion) and enzyme-involved reactions in soil and plants were described in Introduction. To study the isotope fractionation from root to leaves, for example, δ15N values of xylem sap are useful information. To study the isotope fractionation in nitrogen mineralization in soils, for example, δ15N values of nitrate and ammonium produced in soils are useful.

Response: Thanks for your constructive suggestion. We have removed the ‘fractionation’ in the four lines and removed the inappropriate ones elsewhere.

Line 18-19:

but the fertilizer uptake patterns in tea plants (Camellia sinensis) is unclear.

Line 27-29:

The results clarify spatio-temporal nitrogen patterns and isotope compositions of tea leaves treated with different fertilizer types and ensure δ15N tea leaf values can be used to authenticate organic fertilizer methods across different harvest periods and leaf locations.

Line 41-43:

but the spatio-temporal change of nitrogen isotopes in tea plants by different fertilizers is still unclear.

Line 81-84:

This study contributes towards understanding the spatio-temporal dynamic changes of nitrogen stable isotopes in tea plants from different fertilizer types and provides a fundamental basis for the study of nitrogen isotope composition and transfer mechanism for organic tea.

Line 253:

15N fractionation of top’ change into ‘δ15N differences between the top’.

Line 254:

‘Temporal and spatial tea leaf 15N fractionation between top’ change into ‘Temporal and spatial tea leaf δ15N differences (Δ15N) between top’.

Comment 4 [lines 49, 51-67]: Not ‘few studies’ but moderate numbers of studies were reported in Asian tea-growing countries, not only China but more Japan and India. For isotopic fractionations in plant-soil systems, there are many excellent literatures.

Response: Thanks for your constructive suggestion. We modified it and added new reference to clarify the aim of the present study.

Line 53-57:

Previous studies have used different fertilizer types (including fish meal, rapeseed cake and chemical fertilizer) to grow tea plants and found that nitrogen isotope differences of tea leaves correlated with planting year and tea variety [51]. In China, there are only a few tea studies which investigate the nitrogen isotope effects from Chinese commonly used fertilizers.

Line 446-448:

Hayashi, N.; Ujihara, T.; Tanaka, E.; Kishi, Y.; Ogawa, H.; Matsuo, H. Annual variation of natural 15N abundance in tea leaves and its practicality as an organic tea indicator[J]. Journal of agricultural and food chemistry, 2011, 59(18): 10317-10321. https://dx.doi.org/10.1021/jf202215z

Line 73-75:

Researchers have reported that plant δ15N values in plant-soil systems are closely related to soil nitrogen content, nitrogen cycling rate, and plant nitrogen uptake strategy [52].

Line 449-451:

Wang, Y.; Niu, G.; Wang, R.; Rousk, K.; Li, A.; Hasi, M.; Wang, C.; Xue, J.; Yang, G.; Lv, X.; Jiang, Y.; Han, X.; Huang, J. Enhanced foliar 15N enrichment with increasing nitrogen addition rates: Role of plant species and nitrogen compounds[J]. Global Change Biology, 2023, 29(6): 1591-1605. https://doi.org/10.1111/gcb.16555

Comment 5 [lines 77-88]: How much of soils were used in each pot? What are their nitrogen contents and δ15N values? What are the δ15N values of the tea leaves before fertilizer treatments?

Response: Thanks for your constructive suggestion. We used around 3 kg of soil in each pot. Unfortunately, the initial soil composition in each pot was not tested prior to fertilization. However, the d15N values of the untreated soil and leaves are represented by the ‘Control’ samples, where no fertilizer was added, as the ‘Control’ tea potted plants underwent the same sampling and were subjected to the same growing conditions as the fertilized pots. 

Line 94-96:

…contained around 3 kg of soil in each pot. The initial %N and δ15N values of the tea plants and soil are represented by the Control samples. 

Comment 6 [Table 6]: The amounts of fertilizer nitrogen dosed should be reported but not dosed fertilizer amounts.

Response: Thanks for your constructive suggestion. We have adjusted Table 1 to show the actual fertilizer nitrogen dosage instead of the total dosage.

Line 99-101:

Fertilizer type, nitrogen content, dosage and δ15N values used in the experiment are shown in Table 1, with around 1.6g of nitrogen delivered to each pot at the start of the experiment, apart from the Control samples.

Comment 7 [line 122]: ‘isotope fractionation differences’ here is wrong use, only ‘isotope composition differences’, which do not say anything about isotopic fractionation. Isotope fractionations occur during physical and enzyme-involved reactions.

Response: Thanks for your constructive suggestion. We have removed fractionation differences.

Line 137: ‘isotope fractionation differences’ change into ‘isotope composition differences.’

Comment 8 [Results and discussion]: The discussions for isotope fractionations in the plants were only speculations without experimental evidences and important references by scientists working for isotopic fractionations in plants and soils. More information on the time-dependent mineralization from fertilizers and soils as well as δ15N values of mineralized N at plant N uptake is required. The discussions for Table 4 with large SD are over-discussion.

Response: Thanks for your constructive suggestion. The reviewer raises a valid criticism of our work and outlines their desire to see more time-dependent evidence between soil and tea plants. We believe this initial study shows considerable evidence between the uptake and distribution of fertilizer isotopic signatures to different leaf locations to better understand isotopic conservation in space and time. Further work will be undertaken to focus more specifically on the soil-plant transfer in the future. Meanwhile, we added references to help to discuss the results of present study and revised the conclusion parts. We have carefully evaluated this section and made some minor edits to reduce the Table 4 discussion.

Line 179-182:

In previous ecology studies, some papers reported that soil nitrogen composition is the main factor affecting plant δ15N values, and plant leaf δ15N values are strongly related to nitrogen absorption strategies about different nitrogen forms [53-54].

Line 452-458:

Gurmesa, G.; Hobbie, E.; Zhang, S.; Wang, A.; Zhu, W.; Koba, K.; Yoh, M.; Wang, C.; Zhang, Q.; Fang Y. Natural 15N abundance of ammonium and nitrate in soil profiles: New insights into forest ecosystem nitrogen saturation[J]. Ecosphere, 2022, 13(3): e3998. https://doi.org/10.1002/ecs2.3998

Takebayashi, Y.; Koba, K.; Sasaki, Y.; Fang, Y.; Yoh, M. The natural abundance of 15N in plant and soil‐available N indicates a shift of main plant N resources to NO from NH along the N leaching gradient[J]. Rapid Communications in Mass Spectrometry, 2010, 24(7): 1001-1008. https:// 10.1002/rcm.4469

Line 303-311:

Overall, the δ15N signatures of OF and CF treated tea plants are sustained during the early and late spring harvest period, providing clear evidence that the δ15N values of harvested tea from different locations on the tea plant can be directly related to fertilizer type and farming methods. The outcomes of this research show that top and middle tea leaves typically collected during tea harvest can be reliably used to authenticate organic production claims, increasing tea production verification tools and consumer confidence. While our research does not identify the precise internal mechanisms that give rise to these results, further studied using labelled fertilizers will be undertaken to establish kinetic mechanisms and confirm our observations in more detail.

Reviewer 4 Report

Comments and Suggestions for Authors

Review this manuscript:

Manuscript ID: plants-2936224

Title: Fertilizer effect on the nitrogen isotope composition and fractionation between soil and tea plants

Or

Fertilizer spatio-temporal effects on the nitrogen isotope com-2 position of soil and different Camellia sinensis leaf locations

Your results showed that rapeseed fertilizer application changed the δ15N values ​​of tea leaves sampled from different locations on the plant, first showing an enrichment from basal to middle leaves, then a decrease from middle to upper leaves in autumn, early spring and late spring. Δ15N isotopic differences were observed between basal, middle and upper leaves and were clearly influenced by fertilizer type and sampling period.

The outcomes of this research show that  top and middle tea leaves typically collected during tea harvest can be reliably used to  authenticate organic production claims, increasing tea production

Nitrogen isotopes were determined for tea leaves in different spatial positions during different collecting periods. This study contributes towards understanding the spatio-temporal dynamic changes of nitrogen stable isotopes in tea plants from different fertilizer types and provides a fundamental basis for the study  of nitrogen isotope composition and transfer mechanism for organic tea.

Your research is very interesting and worthy of new knowledge. Continue in that direction. It is especially important to set up and introduce organic cultivation into the entire process of green tea production.

I propose to publish the paper in its entirety.

Round 2

Reviewer 3 Report

Comments and Suggestions for Authors

Guo et al. have revised the original manuscript by eliminating all discussions on nitrogen isotope fluctuations in tea plant and soil system and as suggested by Reviewer 1, only variations of nitrogen isotope compositions (δ15N) in three leaf parts and soils from potted tea plants treated with organic fertilizers, urea and no fertilizers are presented. There are already many reports on the isotopic compositions depending on N sources, including the high δ15N values in organic fertilized treatments.

The information obtain in this revised report is useful only locally.

The serious problem in the revised manuscript is misunderstanding included in the main part of Abstract from lines 23 to 27. The results shown in Figure 2 and Table 4 does not support the authors’ statements in the Abstract, which are not precise and not significantly true.

Round 3

Reviewer 3 Report

Comments and Suggestions for Authors

Guo et al. submitted second revised manuscript with more changes in Abstract. The changes increased the number of serious problems in Title and Abstract.

Problem 1 (line 2, Title): The manuscript is only a report from a pot experiment for variations of nitrogen isotope ratios in tea plant and soil. “Fertilizer spatio-temporal effects” should be examined in open fields from the sense of natural isotope abundances.

Problem 2 (lines 22 and 23, Abstract). The δ15N values of top leaves treated with rapeseed cake, are not 3.4‰ to 6.6‰ (see Table 2) and those of soils are not 3.3‰ to 6.0‰ (see Table 3).

Problem 3 (lines 23 to 29, Figure 2): Figure 2 is very small and incomplete for the understanding by readers. The units for “N%” are missed.

Problem 4 (lines 29 to 32): Only results for rapeseed cake treatment are reported in the Abstract. Other organic fertilizer treatments (chick and cow manures) were omitted.
